Misinformation effects in an online sample: results of an experimental study with a five day retention interval

Sievwright Olivia 1 Olivia.Sievwright.1@uni.massey.ac.nz
http://orcid.org/0000-0001-8203-8018 Philipp Michael 1
Drummond Aaron 1
Knapp Katie 2
http://orcid.org/0000-0003-3744-8143 Ross Kirsty 1
1 School of Psychology, Massey University , Palmerston North , New Zealand
2 Work-Learn Institute, University of Waterloo , Waterloo , Canada
Thompson Steven
Electronic publication date: 2021 Nov 18
Publication date: 2021
Volume: 9
Electronic Location ID: e12299
Received 2021 Mar 16; Accepted 2021 Sep 21
Copyright: © 2021 Sievwright et al.
Copyright year: 2021
Copyright holder: Sievwright et al.
License: This is an open access article distributed under the terms of the Creative Commons Attribution License, which permits unrestricted use, distribution, reproduction and adaptation in any medium and for any purpose provided that it is properly attributed. For attribution, the original author(s), title, publication source (PeerJ) and either DOI or URL of the article must be cited.
License URL: https://creativecommons.org/licenses/by/4.0/

Keywords: False memory, Misinformation effect, Online study, ROC analysis, Traumatic memory, Decision making

Funding: School of Psychology at Massey University, New Zealand This research was supported by a Postgraduate Research Award from the School of Psychology at Massey University, New Zealand. The funders had no role in study design, data collection and analysis, decision to publish, or preparation of the manuscript.

==============================
Traditional face-to-face laboratory studies have contributed greatly to our understanding of how misinformation effects develop. However, an area of emerging concern that has been relatively under-researched is the impact of misinformation following exposure to traumatic events that are viewed online. Here we describe a novel method for investigating misinformation effects in an online context. Participants (N = 99) completed the study online. They first watched a 10-min video of a fictional school shooting. Between 5 and 10 days later, they were randomly assigned to receive misinformation or no misinformation about the video before completing a recognition test. Misinformed participants were less accurate at discriminating between misinformation and true statements than control participants. This effect was most strongly supported by ROC analyses (Cohen’s d = 0.59, BF10 = 8.34). Misinformation effects can be established in an online experiment using candid violent viral-style video stimuli.

Introduction

For decades, researchers have demonstrated the harmful effects of misleading post-event information on memory. Numerous laboratory studies have shown that participants exposed to post-event misinformation frequently endorse misinformation as part of their original memory of an event; this is known as the misinformation effect (Loftus, 2005). The application of these findings in legal cases has shed light on the detrimental effects of misinformation in the real world and has contributed to the overturn of wrongful convictions. Misinformation effects may also lead people to remember events as being more traumatic than initially reported (Porter, Taylor & ten Brinke, 2008). This study builds on extant misinformation research by testing a novel method for investigating misinformation effects in an online context.

Misinformation effects are traditionally studied in face-to-face laboratory settings using a three-stage procedure (Loftus, Miller & Burns, 1978). First, participants witness a target event, often through a video depicting a crime. Second, after a delay, participants are exposed to misleading information about the target event—this is often presented through a fictional eyewitness account, fictional news report, memory test, or through discussion with a co-witness. Finally, when tested on their memory for the target event (ranging 5 min to 2 weeks later), participants frequently report misinformation as part of their original recollection. Misinformation effects have led participants to report seeing a thief using a hammer instead of screwdriver (McCloskey & Zaragoza, 1985), or even recall non-existent footage of the car crash in which Diana, Princess of Wales was killed (Ost et al., 2002). This procedure has taught us a great deal about how false memories develop in laboratory conditions (for a review see Loftus, 2005).

One area of emerging concern that has been relatively under-researched is the impact of misinformation following exposure to traumatic events that are viewed online. Advancements in Internet technology now offer us immediate access to coverage of real-world traumatic events (Peterson & Densley, 2017). Consequently, people are inadvertently exposing themselves to footage of such events during their regular Internet use. For example, within 24 h of the 2019 Christchurch, New Zealand mosque attacks, the live-stream of the attack had been uploaded to Facebook over 1.5 million times (Besley & Peters, 2019). This is particularly concerning given accumulating evidence that traumatic content online can have harmful psychological effects for viewers, including post-traumatic stress symptoms, depression, and anxiety (e.g., Pfefferbaum, Nitiéma & Newman, 2019; Redmond et al., 2019; Thompson et al., 2019).

Up-to-the-minute media coverage of unfolding events often includes inaccurate reports based on incomplete information and mistaken details (Rapp & Salovich, 2018; Rich & Zaragoza, 2016). For instance, during the Christchurch mosque attacks it was erroneously reported that ‘a good guy with a gun’ stopped the shooter. In fact the man rushed at the shooter with an electronic payment terminal (Cooke, 2019). Research suggests memory for highly negative events is susceptible to distortion (Nahleen, Strange & Takarangi, 2020; Strange & Takarangi, 2012). Exposure to inaccurate information facilitates misinformation effects for real-world traumatic events witnessed through the media. People who view traumatic videos may be vectors of distressing and inaccurate information to the general public. Thus, memory distortions from misinformation effects may have clinical implications even in the absence of legal ones. For instance, even if people who witness violent online videos are unlikely to be called as factual event witnesses in a trial due to the ability to verify facts from the recorded footage, memory distortions from misinformation effects may have implications for psychological distress and, potentially, the development of clinical disorders such as PTSD. Therefore, it is imperative we understand exactly how misinformation effects work in the context of traumatic online media exposure.

Existing misinformation paradigms are not well-suited to investigating misinformation effects in the context of media exposure to traumatic events. Typical misinformation studies use staged videos of a burglary or car crash that clearly depict fictional events. These videos are often not designed to evoke strong emotional responses. Additionally, the artificial laboratory environment may lead to less intense emotional reactions than would typically occur when witnessing a real-life traumatic event (Chae, 2010). Examining misinformation effects in an online context minimises potential experimenter effects and increases ecological validity by exposing people to traumatic media in the context people are most likely to encounter such material—on their own computers.

Two important methodological decisions need to be made when measuring misinformation effects: retention intervals and type of memory test used. Retention intervals are the period between viewing the target event (encoding) and being exposed to misinformation about the event, and the period between exposure to misinformation and memory retrieval. Although longer retention intervals between encoding and misinformation exposure (i.e., at least 24 h) produce larger misinformation effects (Frost, Ingraham & Wilson, 2002; Paz-Alonso & Goodman, 2008), most studies use retention intervals of 30 min or less. Longer retention intervals between encoding and misinformation exposure have theoretical and pragmatic advantages: theoretically, increasing the retention interval should increase susceptibility to misinformation because event memory decays over time (Conway, 2009; Sekeres et al., 2016). Longer retention intervals also mimic real-world time intervals between witnessing an event and memory retrieval. An online approach should help to minimise practical issues around longer retention intervals because multiple participant visits to the laboratory are not required.

The type of memory test used is important to detecting misinformation effects. Misinformation studies employ different types of memory tests to quantify memory accuracy and misinformation endorsement, including forced-choice recognition, cued recall, free recall, remember/know memory judgements, and source monitoring tests. One of the most common memory tests used in misinformation research is a two-alternative forced-choice recognition test, a test where participants’ answers are categorically coded as correct or incorrect. In these tests, participants are asked about some aspect of the target event and are given two possible response options to choose from. For example, participants might be asked to decide whether the thief’s t-shirt was blue or green. Although a comprehensive analysis of the advantages and disadvantages of the various memory tests is beyond the scope of the present manuscript, there is debate about what type of test is optimal, particularly in the context of eyewitness evidence, and the primary disadvantage of such categorical measures of memory is that they are relatively insensitive to small shifts in discriminability that fail to reach participants’ criteria to change their categorical response (e.g., Wixted & Mickes, 2018).

In this study, we employ a different technique which may be able to detect small discrepancies in people’s ability to distinguish between memory for the original event and memory for post-event misinformation. Rather than asking participants to decide between two competing items, we ask participants to make a categorical judgement about how confident they are that a particular item was present in the target video. For example, participants might be asked how certain they are that the thief’s t-shirt was blue on a scale from 1 (certain this is false) to 6 (certain this is true). These ratings allow us to plot a receiver operating characteristic (ROC) curve, which provides a clearer indication of precisely how memory accuracy differs at varying confidence levels (Stanislaw & Todorov, 1999; Wixted & Mickes, 2018). The advantage of using this type of recognition test is that it may detect subtle changes in memory strength and bias when memory change does not meet the threshold for reporting misinformation as true (Stanislaw & Todorov, 1999). ROC curves enable us to detect reduced certainty that misinformation is false even when accuracy rates are not explicitly affected. For example, misinforming participants that the thief’s t-shirt was blue may cause participants to be less certain that the thief’s t-shirt was actually green. Although they may correctly report ‘false’ for the blue t-shirt, the effect of misinformation can be seen in the reduced certainty ratings.

Present study

This study tests the effectiveness of a novel experimental paradigm for investigating misinformation effects in the context of media exposure to trauma using an ostensibly realistic, online, violent viral-style video as the target event, and a 1-week retention interval. We also used more sensitive estimates of misinformation effects by measuring participants’ confidence in their memory reports to determine discriminability (d’) and points on a ROC curve. To our knowledge, this is the first study to measure misinformation effects after 1 week using such a procedure.

Memory vividness and emotionality ratings have been examined in recent misinformation and trauma therapy studies (e.g., Calvillo & Emami, 2019; Houben et al., 2018; van Schie & Leer, 2019). To contribute to future research, a secondary aim of this study was to establish the minimum therapeutically meaningful change in ratings of memory emotionality and vividness across sessions. Although not elaborated on here, the findings from these analyses will help to inform a baseline of memory vividness and emotionality for a future study.

Materials & methods

All procedures reported here involving human participants were in accordance with the ethical standards of the Massey University Human Ethics Committee (Notification number: 4000021787). Participants were informed of the purpose, procedures, and requirements of the study prior to participating. Informed consent was implied through participation in the study and all participants were debriefed as to the purpose of the study at the conclusion of each study. Preregistration information for this study can be accessed at: https://aspredicted.org/ma4st.pdf. The materials and data for this study are openly available on the Open Science Framework (OSF): https://osf.io/r48bm/?view_only=8137681d7e044d06a6bef57d4604b32f.

Design

This study consisted of two sessions spaced approximately 1 week apart (see Fig. 1). Session 1 was identical for everyone. In Session 2, half of participants were randomly assigned to the misinformation condition and half were assigned to the no-misinformation control condition. Memory vividness and emotionality ratings were compared across a 2(Condition: misinformation or control) × 2(Time: session 1 and session 2) mixed factorial design. We carried out an informal pilot test of the study material with a small group of students prior to collecting data. Feedback from the pilot study indicated the study was clear and plausible.

Figure 1 Schematic representation of the experimental procedure for Session 1 and Session 2.

Participants

Participation was restricted to those who were at least 18 years old. Potential participants were asked not to participate if they had a current or previous diagnosis of post-traumatic stress disorder, depression, or anxiety, or if they had been exposed to a traumatic or violent life event, particularly gun-related violence. We aimed to recruit 100 participants to complete both experimental sessions. This sample size was a compromise between collecting a larger sample size than in previous similar misinformation studies (e.g., Houben et al., 2018) and resource constraints.

One hundred and twenty-nine participants took part in Session 1 of the experiment. Data from 12 participants were excluded for not viewing the entire 10-min video and data from a further 12 participants were excluded for failing both attention checks. Session 2 was made available to the remaining 105 participants 5–10 days after they completed Session 1. Ninety-nine participants completed Session 2. No participants expressed suspicion that incorrect details presented in the misinformation narrative were intentional or part of the study.

The final sample (N = 99) included 53 people (33 male, 20 female) in the misinformation condition and 46 (33 male, 11 female, 2 non-binary) in the control condition. The mean age of participants did not differ between the misinformation condition (M = 28.3, SD = 9.32) and the control condition (M = 26.6, SD = 8.31), t(97) = 0.93, p = 0.36. The mean retention interval between Session 1 and Session 2 also did not differ between the misinformation condition (M = 5.49 days, SD = 1.15) and control condition (M = 5.35 days, SD = 1.06), t(97) = 0.64, p = 0.53.

Measures/materials

Trauma film

In Session 1, participants viewed a 10-min video comprised of excerpts from the film ‘Zero Day’ (Coccio, 2002) to temporarily induce mild feelings of distress similar to those experienced in reaction to witnessing a traumatic event. The film depicts a school shooting and the moments leading up to the shooting using ostensible home-video footage and security camera recordings. The video clip begins with two perpetrators introducing themselves to the camera and planning their attack on the school. The perpetrators are then shown entering the school and killing and tormenting multiple students via security camera recordings. The video ends with the perpetrators taking their own lives as an emergency dispatch officer pleads with them over the phone. An image of playing cards was inserted at the end of the video for attention check purposes, described later.

Vividness rating scale

Memory vividness was measured using a Visual Analogue Scale (VAS) during Session 1 and 1 week later in Session 2. In Session 1, participants rated how vivid (clear) their memory of the trauma film was on a scale from 0 (not vivid at all) to 10 (extremely vivid). At Session 2, participants rated memory vividness using the same scale. They also rated their perceived change in vividness from Session 1 using a 5-point scale with the following response options: (1) a lot more vivid, (2) a little more vivid, (3) the same, (4) a little less vivid, (5) a lot less vivid. We used this to determine the smallest effect size of interest regarding the change in memory vividness over time (Anvari & Lakens, 2019).

Emotionality rating scale

Memory emotionality was also measured using a VAS in Session 1 and 2. Participants rated how emotional their memory for the trauma film was on a scale from 0 (extremely negative) to 10 (extremely positive). At Session 2, participants rated their perceived change in memory emotionality from Session 1 to Session 2 on a 5-point scale, the same as described above for memory vividness.

Attention checks

Two attention checks were included in Session 1 after participants rated the emotionality and vividness of the video. The first attention check was a directed query; participants were presented with a sliding scale and instructed to leave the question blank and not to click on the scale. The second attention check was a multiple-choice question asking participants what image appeared at the end of the trauma film. Response options included: a cartoon gun, playing cards, schoolbag, or penguin. The correct answer was playing cards. Participants who failed these attention checks were excluded from analyses.

Misinformation manipulation

Misinformation was introduced to participants assigned to the misinformation condition in Session 2 through a 447-word fictional eyewitness narrative describing the events depicted in the traumatic video. The narrative contained 12 true statements about the video (e.g., ‘The emergency dispatch lady called out for Andre to pick up the phone’) and eight misinformation statements (e.g., ‘Andre was in the driver’s seat of the car and he was wearing a blue t-shirt’ instead of a camouflage shirt). Participants in the control condition did not receive any misinformation, but instead completed a filler-task for 5 min. In the filler-task, participants searched for thirteen words associated with ice cream flavours that were hidden among a 14 × 14 grid of letters. Participants dragged the computer mouse over the letters that formed a word to solve the word search puzzle. After 5 min, participants were automatically taken to the next task.

Recognition test

Memory accuracy and susceptibility to misinformation was measured in Session 2 using a true/false recognition test with an associated confidence rating for each test item. The test comprised 24 statements about the traumatic video (e.g., “The blonde perpetrator’s name was Cal”, “During the shooting Andre tipped over a desk”), with eight statements directly referring to misinformation details presented in the misinformation narrative (e.g., “The dark-haired perpetrator’s name was Chris” instead of Andre, “The first gun was retrieved from the backseat of the car” instead of boot). The test also contained 4 false items (foil items) unrelated to information presented in the narrative and 12 true statements about the video clip. Each statement was presented individually, and participants were not able to skip items or go back to previous items. Participants indicated how certain they were that each statement was either true or false on a 6-point scale ranging from (1) certain this is false, to (6) certain this is true. Responses of 1–3 for false items were counted as correct and responses of 4–6 for true items were counted as correct (hit). Incorrect answers to misinformation items indicated endorsement of misinformation (false alarm), with a higher number of these questions being incorrect indicating greater susceptibility to misinformation. We measured participants’ ability to discriminate between misinformation items and true items using signal detection and receiver operating characteristic curve analyses, described below.

D-prime (d’)

We used d’ to measure participants’ ability to accurately discriminate between true statements and misinformation statements at test. d’ is derived from the signal detection theory (SDT) of recognition memory, which assumes recognition decisions are based on evidence strength of previously encountered items and new (misinformation or foil) items at test (Stanislaw & Todorov, 1999). Higher values of d’ indicate a greater ability to accurately discriminate between true statements and misinformation statements. If misinformation effects are present in our study, and misinformation participants are poorer at discriminating between true statements and misinformation statements as hypothesised, d’ should be smaller for the misinformation group than for the control group.

Response bias (c)

We used the SDT measure c to compare group differences in response bias. Response bias tells us whether there is a general tendency for participants to respond either “true” or “false” on test items (Stanislaw & Todorov, 1999). In our study, negative values of c represent a bias toward responding “true” and positive values of c represent a bias toward responding “false”. More negative or positive values of c indicate a stronger bias toward responding “true” or “false”, respectively. We expected there would be no difference in overall response bias between the misinformation group and the control group. However, if misinformation effects are present in our study, we would expect the misinformation group to show a greater bias toward responding “true” on misinformation items compared to the control group.

Receiver operating characteristic curves

Receiver operating characteristic (ROC) curves provide a non-parametric and atheoretical estimate of participants’ ability to accurately discriminate between true statements and misinformation statements (Stanislaw & Todorov, 1999). ROC curves are produced by plotting hit rates against false alarm rates for all possible certainty ratings. Chance-level performance, where the hit rate is equal to the false alarm rate, is represented as a diagonal line. This occurs when participants are unable to discriminate between true items and misinformation items and rely on guessing. Good discrimination accuracy is represented by a curve that bows toward the left. The more the ROC curve bows toward the left, the greater the discrimination accuracy. Based on our hypothesis for misinformation effects, we expect the ROC curve for the control group to bow more toward the left than the ROC curve for the misinformation group. We also expected the curves would show similar hit rates for the misinformation and control groups, but the misinformation group would have higher false alarm rates than the control group.

Area under the curve

To quantify discrimination accuracy based on the ROC curves, we calculated AUC for the misinformation and control group ROC curves. AUC can be interpreted intuitively as the proportion of times in which participants correctly discriminate true statements from misinformation statements (Stanislaw & Todorov, 1999). AUC values typically range from 0.5 (chance-level recognition performance) to 1 (perfect recognition performance). Larger AUC indicates greater discrimination accuracy. Consistent with our misinformation effect predictions, we expect the control group will have a larger AUC than the misinformation group, representing poorer discrimination accuracy for the misinformation group.

Procedure

Participants were recruited via the online participant pool Prolific (www.prolific.co), and the experiment was administered in Qualtrics (www.qualtrics.com). The experiment was advertised as a study of memory for traumatic events involving viewing a 10-min video of a fictional school shooting and answering some questions about the video. The study was made available for participants to complete on a desktop computer or laptop, and for those whom English is their first language.

Session 1

Participants began by viewing the trauma video. If they did not watch the full video, they were asked to report the timepoint in which they stopped the video. After the video ended, participants rated the emotionality and vividness of the traumatic video and completed the two attention checks. For exploratory purposes, participants were then invited to comment on their experience of viewing the traumatic video. Finally, participants were thanked for their time, provided contact details for various mental health support services, and reminded they may be asked to participate in another session in 5 days’ time. Session 1 took approximately 20 min to complete.

Session 2

Five days later, participants who completed Session 1 and passed the attention checks were invited to participate in Session 2. Participants were given 5 days to complete Session 2. They again rated the emotionality and vividness of their memory for the traumatic video and indicated their perceived change in emotionality and vividness from Session 1 to Session 2. Those assigned to the misinformation condition read the misinformation narrative, while those in the control condition completed a filler-task for 5 min. Following this, all participants completed a filler-task for a further 5 min before completing the recognition test. At the end of the session, participants were thanked for their time, debriefed as to the purpose of the study, and provided with contact details of mental health support services.

Results

We used the signal detection measure d-prime (d’) and receiver operating characteristic (ROC) curves to assess participants’ ability to correctly distinguish between true statements and misinformation statements 1 week after viewing the traumatic video. We excluded non-misinformation false items from the analysis so we could compare true items directly to misinformation items. A flattening constant was applied to account for floor and ceiling effects for hit rates and false alarm rates. Standard calculations were used for d’, response bias (c), and ROC (Stanislaw & Todorov, 1999). We report 95% confidence intervals for Cohen’s d effect size for each comparison. Post-hoc Bayes’ factors for independent samples t-tests with a default Cauchy prior (0, 0.707) were used to determine the relative evidence in favour of the null or alternative hypothesis for each analysis. Where the assumption of equal variances was violated for between-group comparisons, a Mann–Whitney U test (U) was used.

Results comparing discriminability (d’) for the misinformation and control groups showed a significant difference in correct discrimination between the misinformation (M = 0.21, SD = 0.56) and control group (M = 0.49, SD = 0.75), t(97) = −2.10, p = 0.04, Cohen’s d = 0.42, 95% CI [0.02–0.82]. These results suggest exposure to misinformation 1 week after viewing a traumatic video interfered with participants’ ability to correctly distinguish between true and false statements at test. However, we found contradictory results for the post-hoc BF10 = 1.46; indicating only anecdotal evidence of this effect. We calculated c and AUC to see whether we could further clarify the effect of misinformation on discrimination accuracy.

Group comparisons for c showed participants in the misinformation group (M = 0.79, SD = 0.28) were slightly more biased towards responding “true” compared to control participants (M = 0.55, SD = 0.35), U = 699, p < 0.001, d = 0.76, 95% CI [0.35–1.17]. The moderate-to-large effect size, and post-hoc BF10 = 91.4 indicates very strong evidence of the effect of misinformation on a bias towards responding “true”. There are two possible explanations for this: first, the bias towards responding “true” may reflect misinformation effects; that is, misinformation group participants responded “true” to a greater proportion of misinformation items thereby inflating overall response bias. Second, exposure to misinformation after 1 week may have weakened participants’ memory for the original event, leading them to agree with statements and respond “true” more often than disagreeing and responding “false” across all test items. We further examined response bias using ROC analyses.

In addition to signal detection measures, we used area under the ROC curve as a more sensitive measure of discriminability. We plotted hit rates against false alarm rates for both groups at each level of certainty (from “certain this is false” through to “certain this is true”). Figure 2 displays the ROC curves for the misinformation and control groups. The ROC curves further support the signal detection analyses above, with the control group demonstrating a greater bow to the left, moving further away from chance-level performance, and indicating superior discriminability compared to the misinformation group. The ROC curves also show similar hit rates at each certainty level for the misinformation and control groups. However, the misinformation group show higher false alarm rates than the control group. This suggests the bias toward responding “true” for the misinformation group can more likely be explained by a greater tendency for misinformation participants to respond “true” to misinformation statements than control participants. If it were the case that the misinformation group had an overall tendency to respond “true” across all test items, we would expect to see a greater effect on hit rates.

Figure 2 A comparison of receiver operating characteristic (ROC) curves plotting hit rates against false alarm rates between misinformation and control conditions.

The grey dotted line denotes chance performance, where the hit rate is equivalent to the false alarm rate. Each data point along the curve represents a different certainty level for recognition responses, ranging from “certain this is false” to “certain this is true”.

We calculated AUC for the misinformation and control group ROC curves using the inverse of the trapezoidal rule for AUC. This involved dividing the area above the curves into a series of trapezoids and summing the area of each trapezoid. We subtracted this sum from 1 to give the AUC. An independent samples t-test comparing AUC revealed a significant difference between groups in discrimination accuracy. Participants in the control group correctly determined whether a statement was true or false 77% (SD = 0.12) of the time when true statements and misinformation statements were presented during the recognition test. However, participants in the misinformation group correctly identified true and false items 70% (SD = 0.11) of the time, t(97) = −2.91, p = 0.004, d = 0.59, 95% CI [0.18–0.99]. We found a post-hoc BF10 = 8.34, indicating the data is 8.34 times more likely under the alternative hypothesis (i.e., that participants in the misinformation condition are worse at discriminating between true and false statements) than the null. Although d’ results showed some ambiguity, our AUC comparisons show a clear effect of misinformation on participants’ discrimination accuracy after 1 week.

Exploratory analyses

The analyses in this section were not pre-registered and are therefore exploratory. To explore whether differences in discriminability were due to endorsement of misinformation, we compared hit rates and false alarm rates for both groups. If differences can be attributed to the effects of misinformation, we would expect the misinformation group to have a higher false alarm rate than the control group, but no difference in hit rates between groups. Figure 3 displays the group comparisons of hit and false alarm rates. We found no significant difference in hit rates between the misinformation and control groups, U = 1,146, p = 0.60. However, the false alarm rate was significantly higher for the misinformation group than the control group, U = 659, p < 0.001, d = 0.91. Post-hoc BF10 = 1,052 suggests very strong evidence that between-group differences in discriminability between true and false statements can be attributed to inflated false alarm rates for participants exposed to misinformation.

Figure 3 Recognition test hit and false alarm rates for the misinformation group and control group.

Error bars demonstrate standard error of the mean (SEM) for hit and false alarm rates.

Emotionality and vividness ratings

We used a global transition method (Anvari & Lakens, 2019) to determine the smallest effect size of interest for changes in memory emotionality and vividness for the trauma video from Session 1 to Session 2. We conducted these analyses to determine a cut-off point for the smallest subjectively detectable change in memory vividness and emotionality that is therapeutically meaningful. Emotionality ratings were missing from 3 participants (2 misinformation, 1 control) and vividness ratings were missing from 1 participant in the misinformation group. Data from 96 participants were analysed for emotionality and 98 participants for vividness. Session 1 and Session 2 emotionality ratings were reverse-coded so that higher ratings represented more negative emotionality.

Figure 4 shows the change in emotionality ratings from Session 1 to Session 2 for the two groups. A mixed-model ANOVA established no significant main effect of time, F(1,94) = 0.84, p = 0.36, η2 = 0.003, or condition, F(1,94) = 0.67, p = 0.42, η2 = 0.005. There was also no significant interaction between time and condition for participants’ ratings of emotionality, F(1,94) = 0.19, p = 0.67, η2 = 0.001. This suggests participants’ memory for the school shooting video was equally negative 1 week after viewing the video as it was immediately after viewing the video.

Figure 4 Change in emotionality ratings for the traumatic video from Session 1 to Session 2 for the misinformation and control groups.

Error bars demonstrate standard error of the mean (SEM) for ratings of memory emotionality.

To determine the minimum therapeutically meaningful change in memory emotionality from Session 1 to Session 2, we grouped participants’ responses into three categories: those reporting “no change” (n = 71), “a little change” (n = 23), or “a lot of change” (n = 3). We focused on participants who reported “a little more positive” or “a little more negative” emotionality at session 2. These ratings were combined to form the “little change” category. We conducted a Chi-Square test of independence to examine differences between conditions in the number of participants subjectively reporting “no change”, “a little change”, or “a lot of change” in memory emotionality. No significant differences were found between groups, suggesting no effect of misinformation conditions on perceived change in memory emotionality, χ2(2, N = 97) = 0.81, p = 0.67.

A paired-samples t-test found that, for participants who reported “a little change” in emotionality from Session 1 (M = 6.57, SD = 3.04) to Session 2 (M = 6.24, SD = 1.51), there was a non-significant mean decrease in emotionality of 0.33, t(20) = 0.50, p = 0.63, d = 0.12, 95% CI [−0.32 to 0.54]. This suggests even for those reporting a subjective sense of change, memory emotionality remained stable over time. As a result, we were unable to calculate the smallest therapeutically meaningful effect for emotionality; however, we were able to establish a clear baseline for memory emotionality for the traumatic video over a 1-week period.

Changes in ratings of memory vividness from Session 1 to Session 2 for the misinformation and control groups are displayed in Fig. 5. We found a main effect of time, indicating an overall decrease in vividness for memory of the school shooting video from Session 1 to Session 2, irrespective of experimental condition, F(1,96) = 126.53, p < 0.001, η2 = 0.27. There was also a main effect of condition; participants in the misinformation group tended to rate their memory for the school shooting video as more vivid than participants in the control group F(1,96) = 4.79, p = 0.03, η2 = 0.03. However, the time × condition interaction was non-significant, F(1,96) = 0.82, p = 0.37, η2 = 0.002.

Figure 5 Change in vividness ratings for the traumatic video from Session 1 to Session 2 for the misinformation and control groups.

Error bars demonstrate standard error of the mean (SEM) for ratings of memory vividness.

Again, we grouped participants’ responses into three categories: those reporting “no change” (n = 26), “a little change” (n = 58), or “a lot of change” (n= 15) in memory vividness. A Chi-Square test of independence examined differences between conditions in the number of participants reporting “no change”, “little change”, or “a lot of change” in memory vividness from Session 1 to Session 2. No significant differences were found between the misinformation and control conditions, χ2(2, N = 99) = 1.77, p = 0.41. We focused on participants who reported their memory was “a little more vivid” or “a little less vivid” at Session 2. A paired-samples t-test found that, for participants who reported “little change” in memory vividness from Session 1 (M = 8.10, SD = 1.28) to Session 2 (M = 6.26, SD = 1.15), there was a mean decrease in vividness of 1.84 over time, t(57) = 9.87, p < 0.001, d = 1.30. Memory vividness decreased significantly more for the “little change” group than for the “no change” group, t(82) = −2.19, p = 0.03, d = 0.52. This suggests effect sizes of at least d = 1.30 represent a therapeutically meaningful change in memory vividness over time, with smaller effect sizes representing changes that are too small to be subjectively perceived.

Discussion

To our knowledge, this is the first study to test misinformation effects for a violent viral-style video following a 1-week retention interval using an online platform. We found only anecdotal evidence that misinformation impaired discrimination accuracy when using d’. However, using area under the ROC curve, we established moderate evidence that participants who received misinformation were less accurate when discriminating between true statements and misinformation statements than those who received no misinformation. Our results are consistent with other studies finding an effect of post-event misinformation on memory accuracy and misinformation endorsement (e.g., Loftus et al., 1989; Takarangi, Parker & Garry, 2006).

Our findings highlight the added value of ROC analyses for detecting subtle changes in memory strength, even when misinformation may be below the acceptance threshold. Moreover, the medium effect sizes we found are particularly notable given that our study was conducted online where the capacity for monitoring and control is often much lower than in traditional laboratory experiments.

This study extends our understanding of misinformation effects by using novel stimuli, procedures, and context. We elicited misinformation effects outside of a laboratory using a wholly online experiment and realistic target event to simulate the violent viral-style videos that people may come across online. Many participants commented on the realism of our video, likening it to the 2019 Christchurch mosque shootings and the 1999 Columbine High School massacre. Using an online format, misinformation effects were elicited from a diverse sample that may be more representative of the general population. Moreover, we showed the robustness of misinformation effects under relatively extreme experimental conditions—involving reduced experimental control, a high degree of participant autonomy and trust, and a 1-week retention interval.

We also used more sensitive estimates of misinformation effects by measuring participants’ confidence in their memory reports and using this to determine discriminability (d’) and points on a ROC curve. Signal detection has been used in many previous misinformation studies (e.g., Chan et al., 2017; Paz‐Alonso, Goodman & Ibabe, 2013; van Bergen et al., 2010), however ROC curves are rarely employed in such research. Some argue that a key advantage is that ROC analyses are unaffected by the assumptions about recognition memory that underlie d’ (Wixted & Mickes, 2018). Although the direction of effects in our study was consistent across analyses, ROC analyses provided stronger evidence of misinformation effects. This is because ROC curves enabled us to detect instances where misinformation reduced participants’ certainty in a recognition judgement, even when they did not overtly accept the misinformation. Future research should consider employing ROC analyses as an additional measure to allow for detection of subtle changes in memory strength in misinformation effect research.

We also explored participants’ ratings of vividness and emotionality for their memory of the traumatic video, both immediately after viewing and 1 week later. Irrespective of exposure to misinformation, participants rated their memory for the video as highly vivid and negative. Our results are consistent with other trauma film paradigm studies which have found that experimental analogues to traumatic events can produce intrusive and distressing memories (James et al., 2016). Our findings also contribute to research on the psychological effects of media exposure to traumatic events; although participants in our study were aware the traumatic video was fictional, they appear to have rated their emotions and the vividness of their memories as relatively intense. This provides some indication of the impact of media exposure to real-life traumatic events, such as the live-streamed footage of the Christchurch attacks, which is likely to be even greater.

Interestingly, participants rated their memory as being equally negative 1 week after initially viewing the video but experienced a small decrease in memory vividness over the 1-week period. Our results are partially consistent with findings from similar studies, which have found reductions in memory vividness over time (Houben et al., 2018; van Schie & Leer, 2019). However, these studies also found memory emotionality became more positive over time, which we were unable to replicate. A possible explanation for this is that our video was more distressing than the staged car crash video used in previous studies, thereby we were able to produce more enduring emotional reactions.

Memory vividness tended to be higher for misinformed participants than control participants—across both sessions. According to prominent false memory and recognition decision theories (e.g., fuzzy-trace theory, source monitoring framework, and ballistic accumulator models), poorer memory vividness increases susceptibility to misinformation (Brainerd & Reyna, 2002; Brown & Heathcote, 2008; Johnson, Hashtroudi & Lindsay, 1993). Based on this assumption and our vividness results, we would expect the misinformation group to have better memory performance than the control group. However, we found the opposite; despite perceiving their memory as more vivid, the misinformation group had poorer discrimination accuracy than the control group. Our findings suggest memory vividness did not interfere with misinformation effects and demonstrate that misinformation effects can still occur for memories that are experienced as being relatively vivid.

One limitation of this novel misinformation paradigm is that we cannot control for extraneous variables as well in the online setting compared to a laboratory environment. Distraction during the encoding or testing phases may have affected some participants’ performance on the recognition test. We attempted to minimise such effects by including multiple attention checks throughout the experiment. A second limitation of this study may be the objectively short interval of 5 min between exposure to misinformation and memory retrieval; however, the implications of this are largely unknown. Further studies should investigate whether the effect of misinformation on retrieval is affected by this interval. Additionally, although participants may have incorrectly responded to recognition test items based on their memory for the narrative, rather than the video, we believe this is unlikely for two reasons. First, we provided participants with clear instructions to answer the questions based on their memory for the video. Second, were participants responding based upon the narrative, we would have expected to see a much larger increase in acceptance of the misinformation items than the moderate effects we observed. Lastly, it is also possible that some participants searched YouTube for the target video between sessions or took notes during initial viewing since we told participants they would be asked questions about the video. However, this seems unlikely in our study, given that no participants achieved recognition accuracy greater than 91.7% at test. Furthermore, the fact we still found significant effects in our study highlights the robustness of misinformation effects.

Conclusions

This study tested the viability of a novel method for investigating misinformation effects in an online context. To our knowledge, this is the first study to investigate misinformation effects after 1 week using a fully online sample with ostensibly realistic, viral-style, traumatic stimuli. Using signal detection and area under the receiver operating characteristic curve, results showed participants exposed to misinformation 1 week after viewing a traumatic video were significantly worse at discriminating between true information and misinformation compared to participants not exposed to misinformation. Our results suggest misinformation effects can be established in an online experiment using candid violent viral-style video stimuli. This study also provides evidence for the validity of ROC analyses in misinformation research. We hope our novel materials can further contribute to understandings of memory, misinformation effects, and media exposure to trauma at a faster pace and with more diverse samples than previously achieved.

Supplemental Information

Supplemental Information 1 Memory vividness and emotionality ratings raw data.

Click here for additional data file.

Supplemental Information 2 Misinformation/signal detection/ROC raw data.

Click here for additional data file.

Supplemental Information 3 Data key for misinformation analyses and memory vividness and emotionality analyses.

Click here for additional data file.

We wish to thank Malcolm Louden for technical assistance in the design of this study.

Additional Information and Declarations

Competing Interests

Author Contributions

Human Ethics

Data Availability

The authors declare that they have no competing interests.

Olivia Sievwright conceived and designed the experiments, performed the experiments, analyzed the data, prepared figures and/or tables, authored or reviewed drafts of the paper, and approved the final draft.

Michael Philipp conceived and designed the experiments, authored or reviewed drafts of the paper, and approved the final draft.

Aaron Drummond conceived and designed the experiments, authored or reviewed drafts of the paper, and approved the final draft.

Katie Knapp conceived and designed the experiments, authored or reviewed drafts of the paper, and approved the final draft.

Kirsty Ross conceived and designed the experiments, authored or reviewed drafts of the paper, and approved the final draft.

The following information was supplied relating to ethical approvals (i.e., approving body and any reference numbers):

This study was deemed low-risk research under the Massey University Code of Ethical Conduct. All procedures reported in the manuscript involving human participants were in accordance with the ethical standards of the Massey University’s Human Ethics Committee (Massey University Human Ethics Notification number: 4000021787).

The following information was supplied regarding data availability:

Data is available on the Open Science Framework (OSF): Olivia Sievwright, Michael C Philipp, Aaron Drummond, Katie Knapp, and Kirsty Ross. 2021. “Misinformation Effects in an Online Sample: Results of an Experimental Study with a Five Day Retention Interval.” OSF. September 28. osf.io/r48bm.

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
