# Peer review of "Misinformation effects in an online sample: results of an experimental study with a five day retention interval"

_PeerJ, doi:10.7717/peerj.12299_

## Round 0.1 · original submission · Minor Revisions

Please consider comments from both peer reviewers in revision and resubmission for further editorial review, feedback and prospective publication.

·

Basic reporting

The manuscript is well written and professionally presented. I have only a few minor points where phrasing could be clarified. I present these in order of appearance in the manuscript:
1) Opening sentence (lines 47-48) is a little misleading as written. The 69% figure from the Innocence Project refers to mistaken eyewitness identification (e.g., lineup/showup IDs), which isn’t really relevant to the misinformation effect. To my knowledge, we have no way to quantify the impact of misinformation on wrongful convictions based on our current data
2) The authors describe the typical misinformation paradigm involving misinformation read in the form of a narrative, ostensibly from another witness (lines 58-59). I’m not so sure that this is the predominant method – paradigms that involve discussion between the participant and a co-witness (real or a confederate) are also very common, as are having participants read fictional news reports or similar. Perhaps this section could be softened slightly and other methods of introducing misinformation could be acknowledged
3) The authors define “retention interval” as being between encoding the event and being exposed to misinformation (lines 96-98). However, there are two important retention intervals in any misinformation study – between encoding and misinformation exposure, and between misinformation exposure and memory retrieval. Perhaps the authors could make that clearer and consider the implications of having such a short retention interval between the misinformation and retrieval?
4) On lines 107-113, the authors begin to outline different memory tests that can be used to examine misinformation effects. This paragraph felt incomplete – the authors begin by stating that the type of test is important; they then point to one specific test (2AFC), before moving on. I was expecting more consideration here of different types of memory test and their advantages and disadvantages (e.g., old/new recognition; modified 2AFC (a la Zaragoza et al); source monitoring tests; free recall tests….
5) On line 238, the authors explain that SDT assumes that decisions are based on familiarity. However, SDT is silent as to the evidence that makes up this decision dimension – it could be familiarity, or recollection, or something else. A more agnostic label, such as ‘evidence strength’ might be better here
I was able to access and examine the data files, which included each participant’s raw data and final outcome measures. I was also able to duplicate all of the confirmatory analyses that were reported between line 311 and 357. However, to aid reproducibility further, I have the following recommendations:
1) Please produce a data key to accompany the datasets, which clear specifies what each variable means and (for any categorical variables), how the variables are coded
2) Please clarify whether a flattening constant was applied to deal with any hit rates or false alarm rates at floor/ceiling. From looking at the data file, I think this is the case, but it is not explicitly noted in the manuscript

Experimental design

On the justification of the research:
1. The authors’ case for their paradigm seems to rest on the argument that people now frequently encounter video footage of real criminal events online (e.g., through livestreaming, or because onlookers post video footage to social media), and that they may also encounter misinformation through similar means, or through erroneous news reporting in the confusion that often surrounds unfolding events. However, a person watching such an event is not going to become a key witness - they can’t provide any information over and above what investigators can already see in the video footage. Videotaped incidents are typical in this field (for ethical and logistical reasons), so this isn’t a criticism of using a videotaped event, more as of the mismatch between the rationale that is presented and the applied reality
2. The 1-week delay that is mentioned in the title and abstract isn’t quite accurate – the modal delay appears to be 5 days. In the methods, it becomes apparent that participants completed the memory test 5-10 days after encoding; perhaps the authors could amend the title and abstract to be more accurate?
3. The authors have a secondary question, which involves examining changes in vividness and emotionality over time, and how they relate to perceived changes. I struggled to understand the rationale for these analyses - the authors very briefly state that these analyses were to “establish the minimum therapeutically meaningful change in ratings of vividness and emotionality across sessions”. More detail and justification is needed here. Why is this important? What does minimum therapeutically meaningful change mean, and how is it measured? I also wondered whether this information would be better suited to online supplementary materials
The methods were clearly described, aside from some minor things:
1) Please provide a sample size justification – this needn’t be based on a formal power analysis; it may be based on resource constraints, for example; see Lakens’ recent work on sample size justification: https://psyarxiv.com/9d3yf/
2) Please provide a clearer breakdown of participant exclusions (i.e., number excluded for failing attention checks; number excluded for not watching the video in its entirety)
3) Related to the point above, were participants failed for excluding either attention check, or only if they failed both?
4) On lines 218-228, the authors describe the composition of their memory test, but the numbers don’t seem to add up. From examining the datafile, it looks like there were 12 true statements, but the authors state that there were 8 in the text. I think this is an error – could the authors check?
Overall, the methods appear to be rigorous and are described clearly enough (when combined with the material on the OSF) for a direct replication

Validity of the findings

The authors’ interpretations appear to be sound, and they follow logically from the analyses that were reported. There are some issues that the authors could perhaps consider in their Discussion, though:
1) Demand characteristics. The participants in this study weren’t warned that they had been exposed to misinformation – and we know that warnings reduce the misinformation effect by about half (Blank & Launay, 2014). This might be because it causes participants to engage in more careful source monitoring and/or because they realise that they are not supposed to just ‘go along with’ the suggestions of the experimenter. To what extent might participants have rated critical items as being truthful because they remembered them being in the narrative and they didn’t appreciate/understand that they weren’t supposed to report information from the narrative?
2) Related to the above, did any participants express suspicion in their open ended comments about the misinformation? Loftus has conducted some work on whether salient pieces of misinformation serve as ‘red flags’ which then act as a prophylactic against further bits of misinformation. Was any pilot testing of the materials done to make sure the misinformation was perceived as plausible and not too obvious?

Additional comments

This manuscript describes a single online experiment in which 99 participants viewed a video of a fictional high school shooting (taken from a movie), which was shot to look very realistic (e.g., from handheld video camera and security camera footage). 5-10 days later, the participants were randomly allocated to a control or misinformation condition; participants in the misinformation read a narrative, ostensibly written by another witness (who had seen the same video), which included 8 pieces of misinformation. The participants in the control condition took part in a filler task instead. All of the participants then rated 24 statements about the video on a 1 (false) to 6 (true) scale. Half of the statements were true, 8 were critical misinformation items, and 4 were incorrect filler items. The researchers found evidence of significant misinformation effects, which were strongest when measured using area under the ROC.
The study is clearly reported, appears to have been conducted rigorously, and has reproducible analyses. There are some issues with rationale/justification that could be addressed in a revision, as well as some very minor issues in the Introduction and Methods that could be clarified.

Reviewer 2 ·

Basic reporting

This article is beautifully written and was a pleasure to read. Results are clear and accompanied by appropriate figures and tables. Exploratory analyses are clearly marked as such and there are no concerns regarding these.

The literature review is good but one minor point is that the authors might consider checking whether recent publications from Melanie Takarangi's lab are relevant. Some earlier work from this lab is already cited in the manuscript and I am aware that this group has continued working on misinformation effects in traumatic memories, e.g. Nahleen et al 2020 Does emotional or repeated misinformation increase memory distortion for a trauma analogue event?

Experimental design

The authors have provided a clear overview of relevant literature and a good justification for conducting this research. The design makes use of a well-established method for making responses in signal detection experiments (ratings on a scale from 'certain false' to 'certain true') and applies this nicely to a misinformation paradigm.

One issue the authors may wish to clarify is the description of these responses as a 'categorical judgment' in line 117. It seems this is more akin to a continuous variable than a categorical one.

Validity of the findings

The SDT analyses including ROC, d' and bias are appropriate for the data and are described clearly. The results are clearly reported and the authors have made excellent use of Bayes factors and effect size estimates to give a comprehensive explanation of results. For completeness, comparisons of responses to true vs non-misinformation false items and misinformation items vs non-misinformation false items could be included in supplementary materials but this is not absolutely necessary and can be done at the authors discretion if they choose.

---

## Round 0.2 · accepted · Accept

Congratulations on the acceptance of your peer-reviewed paper to PeerJ.

Reviewer 2 ·

Basic reporting

The authors have addressed the issues regarding literature review raised in my earlier review. I am satisfied the relevant existing findings are covered in their introduction.

Experimental design

The authors have now clarified the measurement of categorical responses.

Validity of the findings

There were no issues to address raised in my earlier review.

Additional comments

The authors have addressed the minor issues I raised in my earlier review. I think this manuscript is now ready for publication and I think it will make a nice contribution to the literature.